# Optical Coherence Tomography Angiography in Type 1 Diabetes Mellitus. Report 4: Glycated Haemoglobin

**DOI:** 10.3390/diagnostics11091537

**Published:** 2021-08-25

**Authors:** Carolina Bernal-Morales, Aníbal Alé-Chilet, Ruben Martín-Pinardel, Marina Barraso, Teresa Hernández, Cristian Oliva, Irene Vinagre, Emilio Ortega, Marc Figueras-Roca, Anna Sala-Puigdollers, Marga Gimenez, Enric Esmatjes, Alfredo Adán, Javier Zarranz-Ventura

**Affiliations:** 1Institut Clínic d’Oftalmologia (ICOF), Hospital Clínic, 08028 Barcelona, Spain; carolbernalmo@gmail.com (C.B.-M.); anibalale@gmail.com (A.A.-C.); marinabarraso@gmail.com (M.B.); tessa.hrndz@gmail.com (T.H.); cristianolpa10@gmail.com (C.O.); mafiguer@clinic.cat (M.F.-R.); ansala@clinic.cat (A.S.-P.); amadan@clinic.cat (A.A.); 2August Pi i Sunyer Biomedical Research Institute (IDIBAPS), 08036 Barcelona, Spain; rbnmartinpinardel@gmail.com (R.M.-P.); ivinagre@clinic.cat (I.V.); eortega1@clinic.cat (E.O.); gimenez@clinic.cat (M.G.); esmatjes@clinic.cat (E.E.); 3Institut Clínic de Malalties Digestives i Metabòliques (ICMDM), Hospital Clínic, 08036 Barcelona, Spain; 4Diabetes Unit, Hospital Clínic, 08036 Barcelona, Spain; 5Centro de Investigación Biomédica en Red de la Fisiopatología de la Obesidad y Nutrición (CIBEROBN), 08036 Barcelona, Spain

**Keywords:** diabetic retinopathy, glycated haemoglobin, HbA1c, oculomics, vessel density, perfusion density, foveal avascular zone, macular thickness, optical coherence tomography, optical coherence tomography angiography

## Abstract

The purpose of this study was to evaluate specifically the relationship between glycated haemoglobin (HbA1c) levels and retinal optical coherence tomography (OCT) and OCT angiography (OCTA) parameters in type 1 Diabetes Mellitus (DM). A total of 478 type 1 DM patients and 115 controls were included in a prospective OCTA trial (ClinicalTrials.gov NCT03422965). Subgroup analysis was performed for controls, no diabetic retinopathy (DM-no DR) and DR patients (DM-DR), and HbA1c levels. OCT and OCTA measurements were compared with HbA1c levels (current and previous 5 years). DM-no DR patients with HbA1c levels >7.5% showed lower VD than DM-DR and controls (20.16 vs. 20.22 vs. 20.71, *p* < 0.05), and showed a significant correlation between HbA1c levels and FAZc (*p* = 0.04), after adjusting for age, gender, signal strength index, axial length, and DM disease duration. DM-DR patients with HbA1c > 7.5% presented greater CRT than DM-no DR and controls (270.8 vs. 260 vs. 251.1, *p* < 0.05) and showed a significant correlation between HbA1c and CRT (*p* = 0.03). In conclusion, greater levels of HbA1c are associated with OCTA changes in DM-no DR patients, and with structural OCT changes in DM-DR patients. The combination of OCTA and OCT measurements and HbA1c levels may be helpful to identify patients at risk of progression to greater stages of the diabetic microvascular disease.

## 1. Introduction

The area of research dedicated to the identification of ocular biomarkers of systemic disease in retinal imaging exams, a field called Oculomics, [1] has raised significant interest in recent years in the study of Alzheimer’s disease [2,3], dementia [4], and cardiovascular diseases [5,6,7,8,9], among which stands prominently Diabetes Mellitus (DM) [10]. While most previous work done in this field leverages fundus retinographies or optical coherence tomography (OCT) images to assess these relationships with systemic diseases, no previous efforts have been implemented on the rich granular data afforded by optical coherence tomography angiography (OCTA) images. OCTA is a newly developed, non-invasive, retinal imaging technique that allows objective quantification of microvascular parameters in the perifoveal vascular network, such as vessel density or flow impairment areas [11,12]. Since this technique allows direct noninvasive in vivo visualization of the microvascular circulation, in the scenario of systemic diseases such as DM it is sensible to think that the detection of microvascular changes at the retinal level may reflect those occurring elsewhere in the body. Similarly, it seems interesting to investigate whether the OCTA-derived parameters present associations with other systemic markers of DM disease, such as the kidney function tests or blood parameters.

Glycated haemoglobin (HbA1c) has been the key measure of glycemic control in diabetic patients for the last 20 years. The Diabetes Control and Complications Trial (DCTT) [13] and the United Kingdom Prospective Diabetes Study (UKPDS) [14] demonstrated that intensive glycemic control with lower levels of HbA1c was proven effective in decreasing the incidence rate of development and progression of diabetic retinopathy (DR) in type 1 and type 2 DM. A threshold of >6.5% has been recommended as one of the DM diagnostic criteria [15,16], and its levels have been closely related to the risk of chronic complications by prospective studies, indicating that this risk increases substantially as the values increase [17,18]. The American Diabetes Association´s recommended goal for HbA1c is <7% [16], and this level is also recommended for prevention of cardiovascular disease in DM patients [19].

The relationship between HbA1c levels and ocular parameters has been investigated with several retinal imaging techniques with controversial results. Recent studies have reported that the application of artificial intelligence algorithms in labelled fundus retinographies from DR screening program datasets provide accurate estimations of blood HbA1c levels [20,21], with significant limitations such as poor external validity in independent cohorts outside each study dataset. Some OCT studies have described positive correlations between HbA1c levels and macular thickness and volume [22,23], and others have suggested negative correlations with choroidal thickness [24,25]. Finally, there is scarce data about HbA1c levels and OCTA-derived parameters [26], this being an area that merits further research.

The purpose of this specific report is to study potential associations between OCTA metrics and HbA1c levels in a large cohort of type 1 DM patients and controls. Subgroup analysis will be performed to evaluate the influence of DR in this relationship, and to investigate further possible correlations between OCTA parameters and HbA1c levels. Finally, the impact of HbA1c variability or progression during the previous 5 years on OCTA measurements will also be explored, to inform whether these features could have direct implications in the systemic management and prognosis of these patients.

## 2. Materials and Methods

### 2.1. Study Design & Study Protocol

The study is cross-sectional and exploratory, with a large cohort of type 1 DM patients recruited from the Diabetes Unit of Hospital Clinic, prospective collection of OCTA images, and ocular and systemic clinical data. The study protocol has been described elsewhere [27]. This project was approved by the Hospital Clinic of Barcelona Institutional Review Board (study protocol version 0.2, 23 November 2016) and registered in the Clinical Trials website (ClinicalTrials.gov NCT03422965). Written informed consent was obtained for all participants. 

### 2.2. Inclusion and Exclusion Criteria

Type 1 Diabetes Mellitus patients undergoing yearly follow up visits as per routine clinical care at the Diabetes Unit of our center were invited to participate and referred for a comprehensive ocular examination in the Ophthalmology department. Controls were collected from healthy volunteers recruited after social media campaigns supported by the Hospital Clinic Communications department. Exclusion criteria included ocular comorbidities (i.e., macular edema, previous ocular surgery, macular laser, intravitreal therapies, etc.), media opacities, or inability to perform complete ocular examinations or provide written informed consent. 

### 2.3. Ocular and Systemic Data

Ocular data collected included best-corrected visual acuity (BCVA), slit-lamp biomicroscopy, intraocular pressure measurement, retinal fundus exam and biometry (IOL Master, Carl Zeiss Meditec, Dublin, CA, USA). DR stage was graded using the International Scale [28]. A comprehensive battery of OCT and OCTA images was performed as described below. Systemic data collected included general characteristics (i.e., age, sex, smoking habit, systolic and diastolic blood pressure, blood hypertension, body mass index) and DM-related characteristics (i.e., DM duration, macrovascular complications, insulin requirements, etc.).

### 2.4. Glycated Haemoglobin Measurements and Definitions

Glycated haemoglobin levels were collected during routine clinical care at the timepoint of the ocular examination and OCTA imaging (2017) and presented as a percentage (%). For the HbA1c variability and progression analysis, historical data was collected yearly from the previous 5 years from electronic medical records (2013–2017). Variability through years was computed by the standard deviation (SD) between measurements, and “high variability” and “low variability” groups were defined by computing the median of each group and using it as threshold. Progression was computed as change from first to final year of HbA1c measurements and classified as “positive” or “negative” depending on this change. 

### 2.5. Structural OCT and OCTA Imaging Protocols

All OCT and OCTA images were obtained using a Cirrus 5000 HD-OCT model (Carl Zeiss Meditec, Dublin, CA, USA). Structural OCT scanning protocols included 6 × 6 mm Macular Cube 512 × 128 cube scans, and OCTA scanning protocols included 3 × 3 mm cube scans centered by foveal fixation. OCT and OCTA image quality check was performed and scans with presence of artifacts, segmentation errors, or signal strength index (SSI) < 7 were excluded from analysis. Structural OCT measurements included central retinal thickness (CRT), macular volume (MV), and average macular thickness (AMT). OCTA quantifications were performed by the built-in commercial software AngioPlex Metrix (v2017, Carl Zeiss Meditec, Dublin, CA, USA) in the superficial capillary plexus of the study eyes, defined by the internal limiting membrane and the inner plexiform layer boundaries. OCTA measurements included vessel density (VD), perfusion density (PD) and foveal avascular zone area (FAZa, mm^2^), perimeter (FAZp, mm), and circularity (FAZc, %). No manual adjustments of the segmentation slab were performed.

### 2.6. Statistical Analysis

Quantitative variables were described using the mean, standard deviation (SD), median, and quartiles (Q_1_, Q_3_). Qualitative variables were described through absolute frequencies and percentages. The normality of distributions was assessed with the Shapiro-Wilk test and homogeneity of variances through Levene’s test. ANOVA tests, Kruskal-Wallis tests, and Chi-square tests for group comparisons were used (where appropriate). T-tests and Mann-Whitney U test were used for pairwise comparisons. Adjusted *p*-values were computed through linear regression models adjusted for age, gender, SSI, axial length, and DM disease duration. Correlations were computed and p-values were adjusted for age, gender, SSI, axial length, and DM disease duration. The Bonferroni correction was applied in all pairwise comparisons. For all the tests, *p*-values < 0.05 were considered as statistically significant. The statistical package R Studio (version 4.0.3) was used for the statistical analysis.

## 3. Results

Data from 593 individuals were evaluated, corresponding to 478 type 1 DM patients (956 eyes) and 115 healthy controls (230 eyes). After systemic and ocular exclusion criteria were applied, a total number of 464 patients were included. To avoid risk of bilaterality bias, only one eye per patient was selected (1 patient/1 eye, *n* = 464 eyes). OCTA images with artifacts (*n* = 24) or low quality defined as SSI < 7 were excluded (*n* = 1). For FAZ parameter analysis, eyes with incorrect FAZ delineation by the automated software were excluded (*n* = 41). A consolidated standard of reporting trials (CONSORT)-style flow diagram describing included and excluded patients and eyes in each individual OCTA analysis is presented in Figure 1.

### 3.1. Baseline Characteristics and Study Groups

Baseline characteristics of study patients are described in Table 1. Subgroup analysis was performed and study cohort was divided in controls (*n* = 72), type 1 DM patients with no DR (DM-no DR, *n* = 247) and DM patients with DR (DM-DR, *n* = 145). At baseline, DM-no DR patients were significantly younger than controls and DM-DR patients (38.3 vs. 47.0 vs. 41.1 years, *p* < 0.05), and DM-DR patients presented significantly longer DM duration than DM-no DR patients (25.9 vs. 16.2 years, *p* < 0.05). No significant differences were observed in structural OCT parameters between study groups. At baseline, some OCTA parameters were significantly different in DM-DR patients compared to DM-no DR and controls. VD and PD were reduced in DM-DR patients compared to DM-no DR patients and controls (19.0 vs. 20.2 vs. 20.6, and 0.35 vs. 0.36 vs. 0.37 respectively, both *p* < 0.05). After adjusting for age, sex, scan quality, DM duration, and axial length, VD results were still significant.

### 3.2. HbA1c Analysis by Study Groups

The comparative analysis of HbA1c levels in the study subgroups is presented in Figure 2 and Table 1. The mean actual HbA1c level was significantly lower in controls than DM-no DR and DM-DR (5.37 vs. 7.34 vs. 7.54, *p* < 0.05), but no differences were observed between DM-no DR and DM-DR eyes (*p* = 0.07). The mean 5-previous year HbA1c level was also significantly lower in controls than DM-no DR and DM-DR (5.37 vs. 7.46 vs. 7.79, *p* < 0.05), and was significantly higher in DM-no DR compared to DM-DR patients (*p* < 0.05). All these results were still significant after adjusting for age, sex, scan quality, DM duration, and axial length.

### 3.3. Differences in OCT and OCTA Parameters by HbA1c Levels

Subgroup analysis was performed by DR status and HbA1c levels, and DM patients were classified in three groups (HbA1c < 6.5, 6.5–7.5 and >7.5%), detailed in Table 2. In the structural OCT analysis, DM-DR patients with HbA1c levels > 7.5% presented greater CRT than those with HbA1c 6.5–7.5% and <6.5% levels (*p* = 0.03), and no differences were observed in DM-no DR eyes. In the OCTA analysis, DM-no DR patients with HbA1c < 6.5% presented significantly higher VD than those with 6.5–7.5% or >7.5% (*p* < 0.05) (Figure 3). No significant differences were observed in any other OCTA parameter or in DM-DR patients.

### 3.4. Influence of 5-Years HbA1c Levels Variability and Progression on OCTA Parameters

The HbA1c measurements from the previous 5 years were analyzed, and patients were classified depending on measurements variability (high/low) or progression (positive/negative) (Table 3). No significant differences were observed in any OCTA parameter in patients with high/low HbA1c variability or positive/negative progression in DM-no DR or DM-DR patients. DM-no DR patients with positive progression in the HbA1c levels showed a trend for lower FAZc (*p* = 0.06) adjusting for age, sex, scan quality, DM duration, and axial length.

### 3.5. Correlations between HbA1c Levels and Structural OCT and OCTA Parameters

Correlations were performed between structural OCT (CRT, MV, and AMT), OCTA parameters (VD, PD, FAZa, FAZp, and FAZc) and HbA1c levels, both actual and mean values from the previous 5 years (Figure 4 and Figure 5). In DM-DR eyes, a significant association was observed between CRT and previous 5 years HbA1c (*p* = 0.03), and a trend was observed with actual HbA1c levels (*p* = 0.07). No associations were found in DM-no DR and structural OCT parameters. In the OCTA analysis, a significant association was observed between FAZc and the actual HbA1c level in DM-no DR patients (*p* = 0.04). Moreover, a trend was observed for VD and the mean 5-years HbA1c levels in DM-DR patients (*p* = 0.08). All the regression models were adjusted for age, gender, SSI, axial length, and DM disease duration.

## 4. Discussion

This report specifically describes significant associations between HbA1c levels and OCTA metrics in a large cohort of type 1 DM patients and controls. We have demonstrated that DM patients with no DR and poor glycemic control (defined as HbA1c > 7.5%) present lower VD than eyes with adequate control (<6.5%). Moreover, we describe that in this subgroup the levels of HbA1c show a significant negative correlation with the FAZc, a parameter that has previously been described as an early marker of microvascular damage. Conversely, DM patients with DR presented associations between HbA1c levels and structural OCT measurements. These findings suggest that both techniques are useful in different time courses of the disease. Meanwhile OCTA, a non-invasive, fast, and reliable retinal imaging technique provides objective quantitative data about the microvascular status in early phases, structural OCT reveals changes in more advanced stages, and both ultimately correlate with the systemic control of the disease, represented by HbA1c levels as key measure of glycemic control in DM patients. 

In our study cohort, the levels of HbA1c were significantly higher in DM-DR patients compared to the other study groups, in particular with regards to the mean previous 5-year HbA1c value. These results are consistent with the existing literature that describes the benefit of intensive glycemic control decreasing the risk of chronic complications as DR development and progression [13,14,16,17,18,19]. In the subgroup analysis for HbA1c levels, in DM-no DR patients the benefit of “adequate” glycemic control (defined as HbA1c <6.5%) compared to “poor” glycemic control (defined as HbA1c > 7.5%) revealed significant differences in VD (*p* < 0.05), and a trend was observed for PD (*p* = 0.08). These are relevant findings that suggest that DM patients with no DR that are poorly controlled and present high HbA1c levels may associate lower VD, indicating an ongoing preclinical microvascular impairment before the clinical DR manifestation onset, as claimed by recent studies [29,30]. In DM-DR patients, no differences were observed in OCTA parameters between HbA1c subgroups, but all the values were lower (VD, PD, FAZc) or higher (FAZa, FAZp) than their DM-no DR equivalents, confirming the sensitivity of OCTA to detect microvascular abnormalities [11,12,31].

The study of potential correlations between HbA1c current and historical data and OCTA parameters merits a specific analysis. We observed a significant negative correlation between HbA1c levels and FAZc in DM-no DR patients, reflecting that in these patients higher HbA1c levels associate lower circularity of the foveal avascular zone (*p* = 0.04). This has been described as an adequate parameter to objectively quantify the perifoveal capillary ring disruption, considered one of the earliest signs of microvascular damage, which has also been associated with visual acuity [32,33]. These results suggest that in DM patients without DR an adequate glycemic control appears essential to prevent this perifoveal microvascular impairment progression, which could affect FAZc and ultimately develop clinical DR. Interestingly, the analysis of the previous 5-years mean HbA1c data did not reveal a significant correlation with any OCTA parameter, although a trend was observed for VD in DM-DR patients (*p* = 0.08). Future studies will shed some light on this specific point, as long-term glycemic control is consistently associated with chronic vascular complications in the literature [13,14,18].

The influence of HbA1c changes in the previous 5 years was also investigated. No significant differences were observed in any OCTA parameter for HbA1c variability or progression, however, in DM-no DR patients a trend was observed for previous 5 years HbA1c positive progression and lower FAZc (*p* = 0.06), suggesting that worsening in the HbA1c levels could associate microvascular alterations in this specific subgroup. Extension studies with HbA1c changes evaluated in longer periods will confirm or discard this preliminary data. 

Most of the previous retinal imaging studies that described associations with HbA1c levels were performed in structural OCT, and mainly investigated the relationship with standard retinal thickness measurements. Two studies reported that higher HbA1c levels were correlated with greater CRT and MV, in two relatively small series of patients (*n* = 97, *n* = 165) [22,34]. In the first one, Yeung et al. suggested that this thickening could be a precursor of diabetic macular edema (DME) in a mixed series of type 1 and type 2 DM patients with more than 10 years of DM disease duration, and Subrayan et al. indicated that the parameter that showed the strongest correlation with HbA1c was MV. In a small retrospective series comparing DME vs. non-DME eyes in type 2 DM patients (*n* = 102), Chou et al. estimated that HbA1c levels > 8% were associated to greater CRT and advocated for strict glycemic control to decrease the risk of progression to DME [23]. Consistently with these reports, in our series we also found significantly greater CRT in patients with HbA1c > 7.5% (*p* < 0.05) and a positive association between CRT and HbA1c levels from the previous 5 years (*p* = 0.03), but only in DM-DR patients. Interestingly, no associations were found in DM-no DR eyes, suggesting that this relationship between HbA1c levels and anatomical changes may exist only later in the course of the disease. This idea is supported by the fact that no intergroup differences were observed in CRT or MV between controls, DM-no DR and DM-DR eyes. The differences with other previous reports results may be explained by differences in the study cohorts, as we have exclusively included type 1 DM patients, as well as the sample size (*n* = 593) among other factors (i.e., age, duration of the disease, ethnicity, etc.). Interestingly, we have only found one study that reports an association between an OCTA parameter and HbA1c levels. Golebiewska et al. reported in a small series of type 1 DM children (*n* = 94, mean age = 13.6 years) that higher levels of HbA1c associated lower parafoveal VD [26]. Meanwhile this finding appears in line with some of our results, this study was conducted in pediatric population and the outcomes are therefore not directly comparable. 

The strengths of this study are the large sample size, the specific type of DM patients evaluated (type 1 DM) and the collection of high-quality data, as both the patients and controls were prospectively included in a clinical trial scenario with bloods and a full battery of retinal imaging tests. However, it also presents a series of limitations. First, the commercial version of the OCT device used in this study only allows the measurement of quantitative data in the superficial capillary plexus and not the deep capillary plexus, that some authors have pointed out as the first site of microvascular impairment in DR progression. Meanwhile this limitation can be overcome with the help of research software in future studies, if the extent of vascular damage is less severe in the superficial plexus somehow this adds clinical significance to the results reported in this study. Second, the HbA1c level threshold for “adequate” glycemic control varies in certain DM patients, as other systemic factors such as the kidney function play a role. Therefore, the same HbA1c figure may reflect different status of the disease and consequently the limits for the patient´s classification in the three subgroups appears arbitrary. Finally, the DM duration was significantly longer in DM-DR patients compared to DM-no DR patients, and although the statistical models were adjusted for this factor, caution is required to interpret the differences observed between both subgroups.

## 5. Conclusions

In conclusion, to the best of our knowledge, this is the first study that focuses specifically on HbA1c levels, the key measure of glycemic control, and OCT and OCTA parameters as potential markers of systemic disease status, the latter being especially relevant in early phases of diabetic microangiopathy. Although the results herein reported need to be confirmed in future studies, and in particular in longitudinal series assessing both the changes in OCT and OCTA parameters and HbA1c levels in time, the findings described in this report suggest that these retinal imaging techniques could be implemented in the routine clinical care annual check-ups, as these objective measurements could have direct implications in the systemic management of these patients in the near future.

## Figures and Tables

**Figure 1 diagnostics-11-01537-f001:**
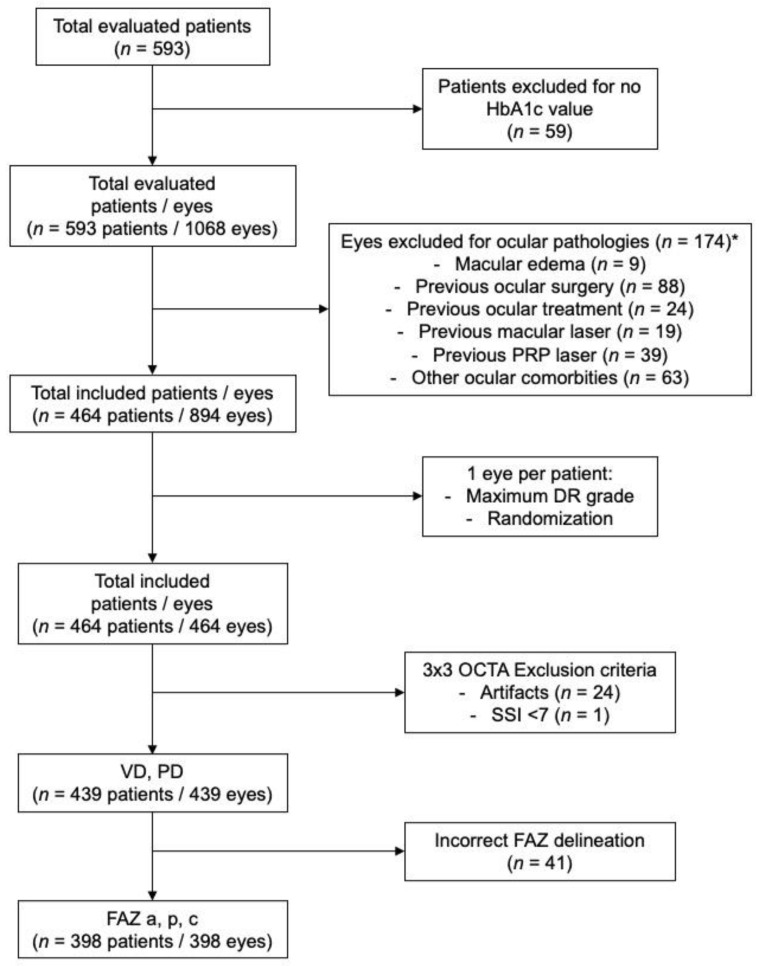
Consolidated standard of reporting trials (CONSORT)-style flow diagram describing included and excluded patients and eyes in each individual OCTA analysis. (* 1 eye = ≥ 1 criteria for exclusion).

**Figure 2 diagnostics-11-01537-f002:**
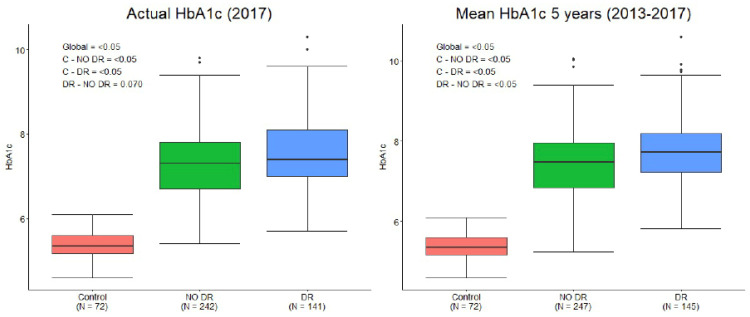
Glycated Haemoglobin (HbA1c) levels in study subgroups. **Left**: Actual HbA1c level at the retinal imaging timepoint (2017). **Right**: Mean HbA1c calculated from previous 5 years timepoints (2013 to 2017).

**Figure 3 diagnostics-11-01537-f003:**
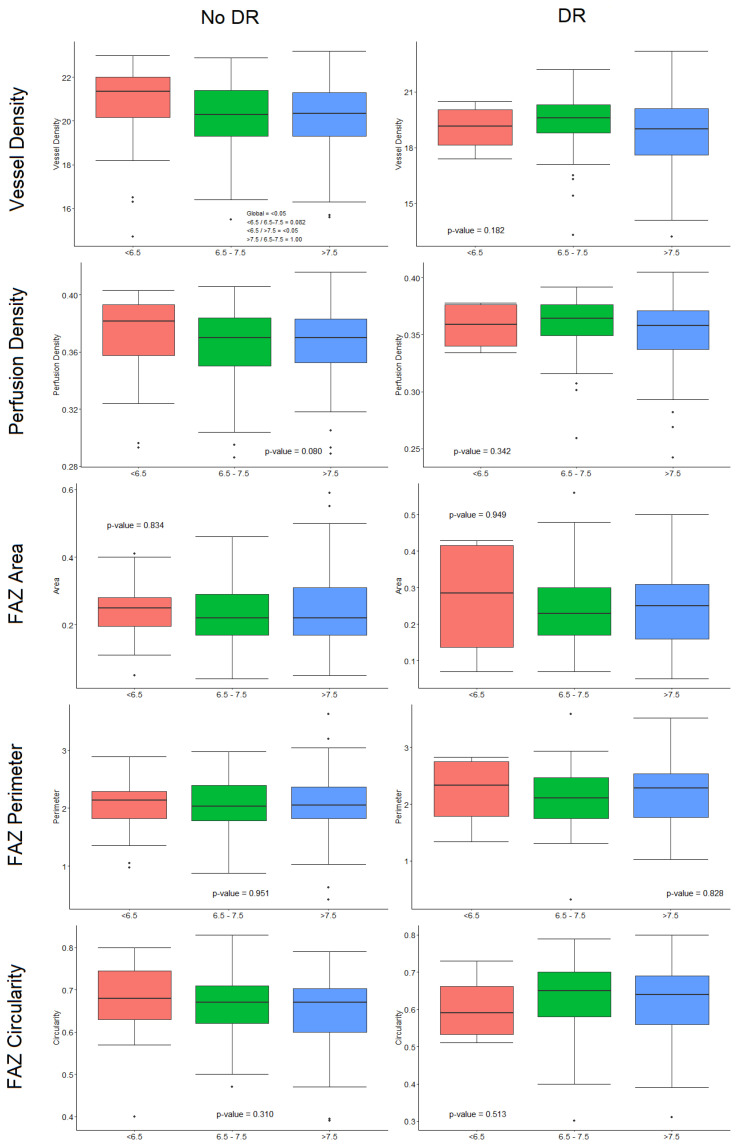
Optical coherence tomography angiography (OCTA) analysis by glycated haemoglobin (HbA1c) level subgroups in diabetes mellitus (DM) patients without and with diabetic retinopathy (DR). **Left**: OCTA parameter analysis in DM-no DR patients. **Right**: OCTA parameter analysis in DM-DR patients. (*p* = values are adjusted by age, sex, signal strength index, diabetes mellitus duration, and axial length).

**Figure 4 diagnostics-11-01537-f004:**
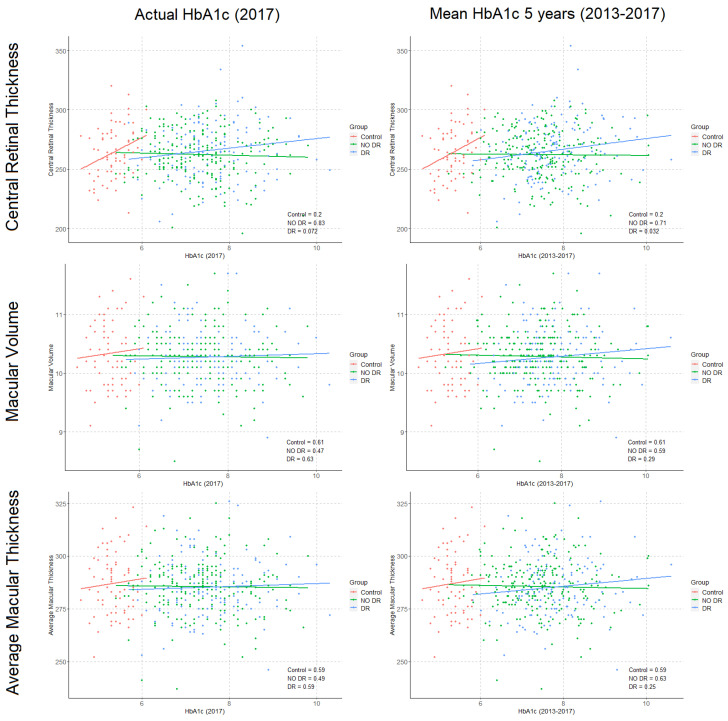
Correlations between glycated haemoglobin (HbA1c) levels and structural optical coherence tomography (OCT) parameters in study subgroups. **Left**: Structural OCT parameters and actual HbA1c level at the retinal imaging timepoint (2017). **Right**: Structural OCT parameters and Mean HbA1c calculated from previous 5 years timepoints (2013 to 2017). (Numerical values represent the *p*-value of correlations, *p*-values are adjusted by age, sex, signal strength index, diabetes mellitus duration, and axial length).

**Figure 5 diagnostics-11-01537-f005:**
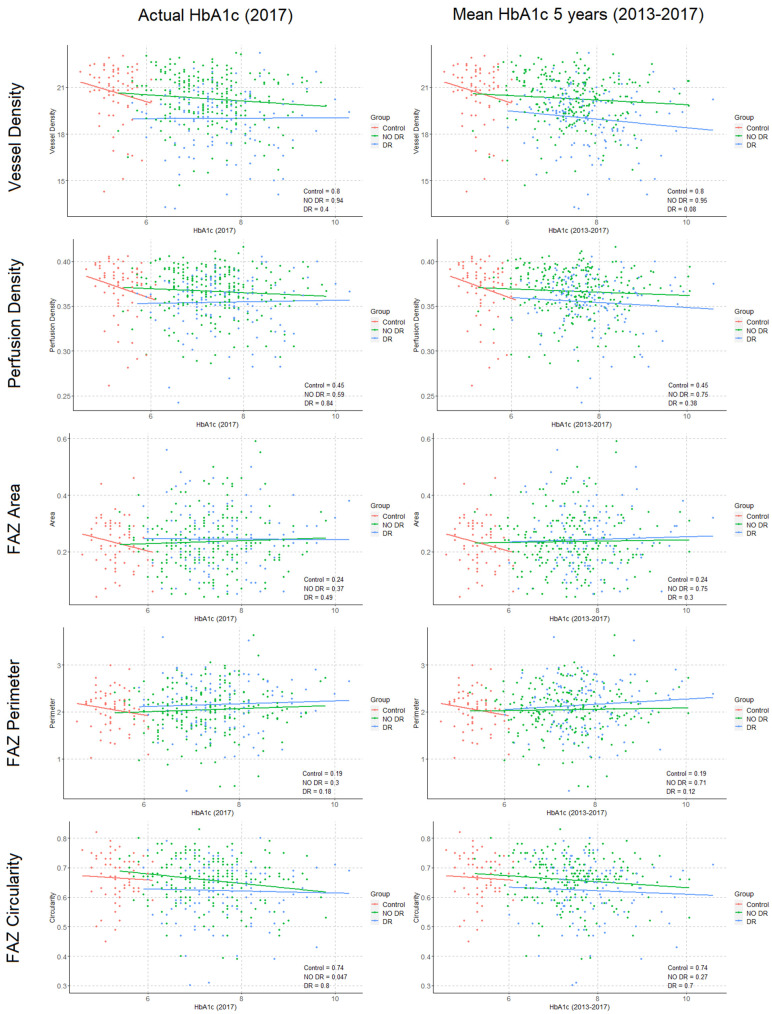
Correlations between glycated haemoglobin (HbA1c) levels and optical coherence tomography angiography (OCTA) parameters in study subgroups. **Left**: OCTA parameters and actual HbA1c level at the retinal imaging timepoint (2017). **Right**: OCTA parameters and Mean HbA1c calculated from previous 5 years timepoints (2013 to 2017). (Numerical values represent the *p*-value of correlations, *p*-values are adjusted by age, sex, signal strength index, diabetes mellitus duration, and axial length).

**Table 1 diagnostics-11-01537-t001:** Demographics and clinical characteristics of study eyes. (* pairwise comparison: No DR vs. DR; *p*-value adjusted by age, sex, scan quality, diabetes mellitus duration and axial length. DR = Diabetic retinopathy, DM = Diabetes mellitus, BMI = Body mass index, SD = Standard deviation, OCT = Optical coherence tomography, OCTA = Optical coherence tomography angiography, FAZ = Foveal avascular zone).

Variable	Number of Eyes (C/No DR/DR)	Statistics	Control	No DR	DR	*p*-Value	*p*-Value Adjusted
**General characteristics**							
Age (years)	(72/247/145)	Mean (SD)	47.04 (14.03)	38.34 (12.55)	41.16 (10.80)	<0.05	-
		Median (Q_1_, Q_3_)	48.65 (34.80, 59.65)	37.10 (27.55, 47.40)	39.30 (33.70, 48.70)		
Sex, female	(72/247/145)	n (%)	47 (65.3%)	131 (53.0%)	70 (48.3%)	0.060	-
Smoking habits	(72/246/145)						
*Non smoker*		n (%)	52 (72.2%)	158 (64.2%)	86 (59.3%)	0.633	-
*Actual smoker*		n (%)	4 (5.6%)	53 (21.5%)	31 (21.4%)	0.284	-
*Ex-smoker*		n (%)	16 (22.2%)	35 (14.2%)	28 (19.3%)	0.595	-
Hypertension	(71/247/145)	n (%)	9 (12.7%)	20 (8.1%)	19 (13.1%)	0.229	-
BMI (kg/m^2^)	(70/244/145)	Mean (SD)	23.54 (3.34)	24.43 (3.72)	25.35 (3.80)	<0.05	-
		Median (Q_1,_ Q_3_)	23.48 (21.17, 25.52)	23.80 (21.66, 26.86)	24.69 (22.72, 27.40)		
**Diabetes-related characteristics**							
DM duration (years)	(0/246/143)	Mean (SD)	0.00 (0.00)	16.29 (9.65)	25.97 (8.97)	**<0.05 ***	-
		Median (Q_1,_ Q_3_)	0.00 (0.00, 0.00)	15.85 (8.43, 21.85)	26.00 (20.35, 32.25)		
Macrovascular complications	(72/246/145)						
*Cerebrovascular disease*		n (%)	0 (0.0%)	1 (0.4%)	3 (2.1%)	0.158	-
*Ischemic heart disease*		n (%)	1 (1.4%)	2 (0.8%)	2 (1.4%)	0.839	-
*Peripheral vascular disease*		n (%)	0 (0.0%)	1 (0.4%)	1 (0.7%)	0.763	-
Insulin requirements (UI/kg)	(0/243/145)	Mean (SD)	0.00 (0.00)	0.62 (0.24)	0.65 (0.24)	0.127 *	-
		Median (Q_1,_ Q_3_)	0.00 (0.00, 0.00)	0.60 (0.45, 0.78)	0.64 (0.51, 0.80)		
HbA1c (2017)	(72/242/141)	Mean (SD)	5.37 (0.33)	7.34 (0.86)	7.54 (0.87)	<0.05	<0.05
		Median (Q_1,_ Q_3_)	5.35 (5.18, 5.60)	7.30 (6.70, 7.80)	7.40 (7.00, 8.10)		
Mean HbA1c (2017–2013)	(72/247/145)	Mean (SD)	5.37 (0.33)	7.46 (0.87)	7.79 (0.83)	<0.05	<0.05
		Median (Q_1,_ Q_3_)	5.35 (5.18, 5.60)	7.48 (6.84, 7.97)	7.72 (7.23, 8.20)		
**Ocular Measurements**							
Visual Acuity	(72/247/145)	Mean (SD)	0.97 (0.06)	0.98 (0.58)	0.93 (0.11)	<0.05	-
		Median (Q_1,_ Q_3_)	1.00 (0.99, 1.00)	0.95 (0.95, 1.00)	0.95 (0.90, 1.00)		
Axial Length (mm)	(72/245/144)	Mean (SD)	23.65 (1.00)	23.65 (1.14)	23.33 (1.17)	<0.05	-
		Median (Q_1,_ Q_3_)	23.48 (22.88, 24.42)	23.53 (22.84, 24.40)	23.19 (22.66, 23.90)		
**OCTA—3 × 3 mm**							
Vessel Density (mm^−1^)	(67/236/136)	Mean (SD)	20.65 (1.87)	20.26 (1.59)	19.00 (1.86)	<0.05	<0.05
		Median (Q_1,_ Q_3_)	21.10 (20.05, 22.00)	20.50 (19.30, 21.40)	19.40 (18.05, 20.20)		
Perfusion Density	(67/236/136)	Mean (SD)	0.370 (0.031)	0.366 (0.026)	0.354 (0.029)	<0.05	0.087
		Median (Q_1,_ Q_3_)	0.377 (0.363, 0.391)	0.371 (0.352, 0.385)	0.362 (0.341, 0.375)		
FAZ Area (mm^2^)	(62/216/120)	Mean (SD)	0.230 (0.086)	0.236 (0.100)	0.243 (0.106)	0.810	0.691
		Median (Q_1,_ Q_3_)	0.225 (0.170, 0.290)	0.225 (0.170, 0.290)	0.230 (0.160, 0.310)		
FAZ Perimeter (mm)	(62/216/120)	Mean (SD)	2.049 (0.409)	2.046 (0.495)	2.150 (0.537)	0.124	0.271
		Median (Q_1,_ Q_3_)	2.085 (1.782, 2.282)	2.060 (1.788, 2.370)	2.200 (1.750, 2.532)		
FAZ Circularity	(62/216/120)	Mean (SD)	0.665 (0.078)	0.659 (0.078)	0.623 (0.097)	<0.05	0.151
		Median (Q_1,_ Q_3_)	0.670 (0.620, 0.720)	0.670 (0.617, 0.710)	0.640 (0.570, 0.690)		
**OCT–Macular Cube**							
Central Macular Thickness (μm)	(69/241/143)	Mean (SD)	264.61 (21.68)	262.26 (20.23)	265.92 (22.92)	0.252	0.489
		Median (Q_1,_ Q_3_)	264.0 (248.0, 280.0)	262.0 (250.0, 276.0)	265.0 (250.0, 281.5)		
Macular Volume	(69/241/143)	Mean (SD)	10.34 (0.54)	10.28 (0.45)	10.27 (0.49)	0.706	0.454
		Median (Q_1,_ Q_3_)	10.3 (9.9, 10.7)	10.3 (10.0, 10.6)	10.2 (9.9, 10.6)		
Average Macular Thickness (μm)	(69/241/143)	Mean (SD)	287.06 (14.83)	285.52 (12.58)	285.29 (13.45)	0.707	0.425
		Median (Q_1,_ Q_3_)	285.0 (276.0, 296.0)	286.0 (279.0, 293.0)	284.0 (276.0, 293.5)		

**Table 2 diagnostics-11-01537-t002:** Subgroup analysis by DR status and HbA1c levels.

Diabetic Retinopathy		Statistics		HbA1c Levels (%)		*p*-Value
			<6.5	6.5–7.5	>7.5	
* **No DR** *	**OCTA parameter**		(Eyes = 32)	(Eyes = 94)	(Eyes = 110)	
	Vessel Density (mm^-1^)	Mean (SD)	20.71 (1.97)	20.22 (1.50)	20.16 (1.53)	<0.05
		Median (Q_1,_ Q_3_)	21.35 (20.18, 22.00)	20.35 (19.30, 21.30)	20.35 (19.30, 21.30)	
	Perfusion Density	Mean (SD)	0.373 (0.029)	0.365 (0.025)	0.366 (0.025)	0.080
		Median (Q_1,_ Q_3_)	0.382 (0.358, 0.393)	0.370 (0.350, 0.384)	0.370 (0.353, 0.383)	
			(Eyes = 31)	(Eyes = 89)	(Eyes = 96)	
	FAZ Area (mm^2^)	Mean (SD)	0.237 (0.081)	0.233 (0.098)	0.238 (0.107)	0.834
		Median (Q_1,_ Q_3_)	0.250 (0.195, 0.280)	0.220 (0.170, 0.290)	0.220 (0.170, 0.310)	
	FAZ Perimeter (mm)	Mean (SD)	2.041 (0.446)	2.059 (0.463)	2.036 (0.541)	0.951
		Median (Q_1,_ Q_3_)	2.140 (1.825, 2.295)	2.030 (1.780, 2.400)	2.055 (1.823, 2.363)	
	FAZ Circularity	Mean (SD)	0.677 (0.082)	0.661 (0.071)	0.651 (0.083)	0.310
		Median (Q_1,_ Q_3_)	0.680 (0.630, 0.745)	0.670 (0.620, 0.710)	0.670 (0.600, 0.703)	
	OCT parameter		(Eyes = 33)	(Eyes = 95)	(Eyes = 113)	
	Macular Central Thickness (μm)	Mean (SD)	261.21 (21.70)	262.68 (17.79)	262.21 (21.83)	0.937
		Median (Q_1,_ Q_3_)	260.0 (246.0, 276.0)	263.0 (251.0, 276.0)	263.0 (248.0, 276.0)	
	Macular Volume	Mean (SD)	10.25 (0.49)	10.26 (0.41)	10.31 (0.48)	0.736
		Median (Q_1,_ Q_3_)	10.3 (10.1, 10.5)	10.3 (10.1, 10.5)	10.3 (10.0, 10.6)	
	Macular Average Thickness (μm)	Mean (SD)	284.52 (13.87)	284.96 (11.29)	286.27 (13.26)	0.799
		Median (Q_1,_ Q_3_)	286.0 (279.0, 292.0)	285.0 (279.0, 291.5)	286.0 (278.0, 294.0)	
* **DR** *	**OCTA parameter**		(Eyes = 4)	(Eyes = 51)	(Eyes = 81)	
	Vessel Density (mm^-1^)	Mean (SD)	19.05 (1.41)	19.31 (1.68)	18.80 (1.97)	0.182
		Median (Q_1,_ Q_3_)	19.15 (18.15, 20.05)	19.60 (18.80, 20.30)	19.00 (17.60, 20.10)	
	Perfusion Density	Mean (SD)	0.358 (0.023)	0.359 (0.026)	0.352 (0.031)	0.342
		Median (Q_1,_ Q_3_)	0.359 (0.340, 0.377)	0.364 (0.349, 0.377)	0.358 (0.337, 0.371)	
			(Eyes = 4)	(Eyes = 45)	(Eyes = 71)	
	FAZ Area (mm^2^)	Mean (SD)	0.268 (0.180)	0.244 (0.108)	0.240 (0.102)	0.949
		Median (Q_1,_ Q_3_)	0.285 (0.138, 0.415)	0.230 (0.170, 0.300)	0.250 (0.160, 0.310)	
	FAZ Perimeter (mm)	Mean (SD)	2.208 (0.701)	2.112 (0.559)	2.171 (0.521)	0.828
		Median (Q_1,_ Q_3_)	2.330 (1.790, 2.748)	2.110 (1.750, 2.470)	2.280 (1.770, 2.535)	
	FAZ Circularity	Mean (SD)	0.605 (0.100)	0.634 (0.098)	0.617 (0.097)	0.513
		Median (Q_1,_ Q_3_)	0.590 (0.533, 0.663)	0.650 (0.580, 0.700)	0.640 (0.560, 0.690)	
	**OCT parameter**		(Eyes = 5)	(Eyes = 56)	(Eyes = 82)	
	Macular Central Thickness (μm)	Mean (SD)	251.20 (29.21)	260.00 (20.81)	270.87 (22.85)	<0.05
		Median (Q_1,_ Q_3_)	253.0 (245.0, 269.0)	261.0 (243.8, 273.5)	270.5 (257.3, 284.8)	
	Macular Volume	Mean (SD)	9.96 (0.22)	10.28 (0.50)	10.29 (0.49)	0.335
		Median (Q_1,_ Q_3_)	9.9 (9.9, 10.0)	10.2 (9.9, 10.6)	10.3 (10.0, 10.6)	
	Macular Average Thickness (μm)	Mean (SD)	276.20 (6.06)	285.36 (13.66)	285.79 (13.54)	0.303
		Median (Q_1,_ Q_3_)	274.0 (274.0, 279.0)	284.0 (277.5, 293.3)	285.0 (277.3, 294.0)	

**Table 3 diagnostics-11-01537-t003:** Influence of 5-years HbA1c variability and progression on Optical coherence tomography angiography (OCTA) parameters. Subgroup analysis by DR status. (*p*-value adjusted by age, sex, scan quality, diabetes mellitus duration, and axial length).

		No DR			DR		
OCTA Parameter	Statistics	High Variability	Low Variability	*p*-Value Adjusted	High Variability	Low Variability	*p*-Value Adjusted
		(Eyes = 107)	(Eyes = 109)		(Eyes = 64)	(Eyes = 63)	
**Vessel Density** (mm^−1^)	Mean (SD)	20.25 (1.57)	20.35 (1.58)	0.905	18.93 (2.00)	19.12 (1.76)	0.635
	Median (Q_1,_ Q_3_)	20.40 (19.30, 21.40)	20.50 (19.60, 21.50)		19.10 (17.80, 20.22)	19.50 (18.40, 20.20)	
**Perfusion Density**	Mean (SD)	0.365 (0.025)	0.369 (0.026)	0.699	0.353 (0.033)	0.357 (0.026)	0.475
	Median (Q_1,_ Q_3_)	0.370 (0.352, 0.383)	0.372 (0.354, 0.389)		0.362 (0.337, 0.376)	0.362 (0.347, 0.374)	
		(Eyes = 96)	(Eyes = 101)		(Eyes = 56)	(Eyes = 57)	
**FAZ Area** (mm^2^)	Mean (SD)	0.24 (0.11)	0.24 (0.10)	0.572	0.25 (0.11)	0.23 (0.10)	0.466
	Median (Q_1,_ Q_3_)	0.23 (0.17, 0.30)	0.23 (0.17, 0.31)		0.26 (0.17, 0.31)	0.23 (0.16, 0.29)	
**FAZ Perimeter** (mm)	Mean (SD)	2.07 (0.47)	2.06 (0.54)	0.861	2.17 (0.62)	2.13 (0.44)	0.998
	Median (Q_1,_ Q_3_)	2.06 (1.85, 2.35)	2.12 (1.77, 2.41)		2.29 (1.72, 2.55)	2.11 (1.77, 2.45)	
**FAZ Circularity**	Mean (SD)	0.66 (0.08)	0.66 (0.08)	0.786	0.63 (0.10)	0.62 (0.09)	0.934
	Median (Q_1,_ Q_3_)	0.67 (0.60, 0.71)	0.67 (0.63, 0.71)		0.66 (0.58, 0.69)	0.63 (0.57, 0.69)	
		**Positive progression**	**Negative progression**		**Positive progression**	**Negative progression**	
		(Eyes = 94)	(Eyes = 107)		(Eyes = 36)	(Eyes = 80)	
**Vessel Density** (mm^−1^)	Mean (SD)	20.31 (1.67)	20.30 (1.48)	0.359	19.46 (1.93)	18.80 (1.82)	0.136
	Median (Q_1,_ Q_3_)	20.70 (19.22, 21.40)	20.40 (19.40, 21.40)		19.70 (18.25, 20.92)	19.30 (17.88, 20.10)	
**Perfusion Density**	Mean (SD)	0.367 (0.027)	0.367 (0.024)	0.185	0.360 (0.029)	0.352 (0.029)	0.146
	Median (Q_1,_ Q_3_)	0.374 (0.353, 0.388)	0.371 (0.353, 0.383)		0.362 (0.347, 0.382)	0.363 (0.341, 0.369)	
		(Eyes = 88)	(Eyes = 95)		(Eyes = 33)	(Eyes = 70)	
**FAZ Area** (mm^2^)	Mean (SD)	0.24 (0.10)	0.24 (0.11)	0.314	0.23 (0.09)	0.25 (0.11)	0.913
	Median (Q_1,_ Q_3_)	0.23 (0.18, 0.31)	0.23 (0.17, 0.29)		0.23 (0.16, 0.29)	0.23 (0.17, 0.32)	
**FAZ Perimeter** (mm)	Mean (SD)	2.09 (0.51)	2.03 (0.51)	0.154	2.11 (0.44)	2.17 (0.58)	0.977
	Median (Q_1,_ Q_3_)	2.09 (1.86, 2.42)	2.11 (1.71, 2.32)		2.14 (1.67, 2.47)	2.25 (1.78, 2.58)	
**FAZ Circularity**	Mean (SD)	0.65 (0.08)	0.67 (0.07)	0.069	0.64 (0.08)	0.62 (0.10)	0.294
	Median (Q_1,_ Q_3_)	0.66 (0.60, 0.70)	0.67 (0.63, 0.71)		0.63 (0.60, 0.68)	0.65 (0.55, 0.69)	

## Data Availability

The datasets used and/or analyzed during the current study are available from the corresponding author on reasonable request.

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
