# Peer review of "Optical Coherence Tomography Angiography in Type 1 Diabetes Mellitus. Report 4: Glycated Haemoglobin"

_diagnostics, 2021, doi:10.3390/diagnostics11091537_

Round 1

Reviewer 1 Report

Overall, the potential prediction probability of OCT/OTCA and HbA1c is reasonable and scientific. The research had a well study design and presentation. However, I have some comments about the paper presentation.

  1. “Abstract : DM-no DR patients with HbA1c levels >7.5% showed lower VD (20.16 vs 20.22 vs 20.71, p<0.05), and showed a significant correlation between HbA1c levels and FAZc 27 (p=0.04), after adjusting for age, gender, signal strength index, axial length and DM disease duration”  should this sentence be described more specifically, like (20.16 in DM-no DR vs 20.22 Control vs 20.71 DM DR, p<0.05), otherwise, readers would very confuse what do the author describe.
  2. Also in this sentence, the groups list should be consistence or should be described “DM-DR patients with HbA1c >7.5% presented greater CRT (270.8 vs 260 vs 251.1, p<0.05) and 29 showed a significant correlation between HbA1c and CRT (p=0.03)”

  1. lack of internal and inter between group p-value for smoking habits 
  2. table 1 legend should included all the abbreviates, like DR:diabetes retinopathy, no DR: no diabetes retinopathy 
  3. Table 1 Axial Length should add unit (mm)
  4. Table 1 Vessel Density (mm-1) , I am not so sure this is a correct unit, should it be Superscript ? mm-1 ?
  5. Table 3 , the variables of adjust P-value should also be described in the table legend.
  6. Figure 4, figure 5, what’s the true meaning of the value described in the figure, does it mean the correlation R2 , or does it mean the p value of correlation ?

Author Response

Overall, the potential prediction probability of OCT/OTCA and HbA1c is reasonable and scientific. The research had a well study design and presentation. However, I have some comments about the paper presentation.

Many thanks to the reviewer for the positive comments.

1. “Abstract : DM-no DR patients with HbA1c levels >7.5% showed lower VD (20.16 vs 20.22 vs 20.71, p<0.05), and showed a significant correlation between HbA1c levels and FAZc 27 (p=0.04), after adjusting for age, gender, signal strength index, axial length and DM disease duration”  should this sentence be described more specifically, like (20.16 in DM-no DR vs 20.22 Control vs 20.71 DM DR, p<0.05), otherwise, readers would very confuse what do the author describe.

The text has been amended as suggested.

2. Also in this sentence, the groups list should be consistence or should be described “DM-DR patients with HbA1c >7.5% presented greater CRT (270.8 vs 260 vs 251.1, p<0.05) and 29 showed a significant correlation between HbA1c and CRT (p=0.03)”

The text has been amended as suggested.

1. lack of internal and inter between group p-value for smoking habits 

The p values have been included as suggested.

2. table 1 legend should included all the abbreviates, like DR:diabetes retinopathy, no DR: no diabetes retinopathy 

Done.

3. Table 1 Axial Length should add unit (mm)

Done.

4. Table 1 Vessel Density (mm-1) , I am not so sure this is a correct unit, should it be Superscript ? mm-1 ?

Many thanks for spotting this typo, it has been corrected now.

5. Table 3, the variables of adjust P-value should also be described in the table legend.

Done

6. Figure 4, figure 5, what’s the true meaning of the value described in the figure, does it mean the correlation R2 , or does it mean the p value of correlation ?

The value means the p value of correlation, apologies for not being clear enough in the first submission. We have included now a sentence in the Figure caption to clarify this in both figures, many thanks for highlighting this detail.

Reviewer 2 Report

Revision of the manuscript entitled:

Optical Coherence Tomography Angiography in type 1 Diabetes Mellitus. Report 4: Glycated Haemoglobin

Summary

Very interesting manuscript on the association of the metabolic control of diabetes mellitus and the findings in the OCTA

Strengths:

The greatest strength is the number of patients with type 1 diabetes mellitus included with a 5-year follow-up.

The authors include in the text a large amount of data obtained through tables and figures, analyzing with a good methodology all the parameters that can be obtained from the OCTAC. Commentaries

  1. The authors study metabolic control through HbA1c levels, but another important risk factor for arterial hypertension is scarcely studied. Only Table 1 makes a reference to the presence of arterial hypertension and the presence of diabetic retinopathy.

I believed that the authors should expand on this point and indicate:

If they have taken into account, the presence of arterial hypertension with good or bad control and relate it to the changes observed in the OCTA

A second point is whether they have taken into account separately the systolic and diastolic tension levels with these changes. The authors do not take into account some data that may explain the failure to attend the appointment.

  1. Another factor that can affect retinal vascularization is renal status. Have the authors taken into account renal function (through glomerular filtration or the presence of albumin in urine) with the findings in the OCTA?

Minor commentaries

  1. The authors base the study on eyes and not patients, which is correct, but could indicate how many patients are studied in each section of Figure 1
  2. In the introduction the authors make the following sentence: “The advent of artificial intelligence applications in ophthalmology has made possible to estimate blood HbA1c levels from fundus retinographies based on existing large datasets from DR screening programs,…”

I think that since it is not proven, they should change it since it implies that through a retinography we can determine the value of HbA1c which is not true

Resume.

In conclusion, an interesting study but with some questions that the authors should answered

Author Response

Revision of the manuscript entitled: 

Optical Coherence Tomography Angiography in type 1 Diabetes Mellitus. Report 4: Glycated Haemoglobin

Summary

Very interesting manuscript on the association of the metabolic control of diabetes mellitus and the findings in the OCTA

Strengths:

The greatest strength is the number of patients with type 1 diabetes mellitus included with a 5-year follow-up.

The authors include in the text a large amount of data obtained through tables and figures, analyzing with a good methodology all the parameters that can be obtained from the OCTA.

Many thanks for the positive comments.

Commentaries

1. The authors study metabolic control through HbA1c levels, but another important risk factor for arterial hypertension is scarcely studied. Only Table 1 makes a reference to the presence of arterial hypertension and the presence of diabetic retinopathy.

I believed that the authors should expand on this point and indicate:

If they have taken into account, the presence of arterial hypertension with good or bad control and relate it to the changes observed in the OCTA.

Many thanks for the important point raised. As part of the clinical examinations performed in the prospective trial, we collected data about all known cardiovascular risk factors including arterial hypertension (systolic and diastolic), lipid profile including HDL, LDL and VLDL cholesterol levels, kidney function tests and data about systemic treatments, such as antihypertensive drugs, fibrates or diuretics (as well as the daily doses of each drug). Of all these factors, for the purposes of clarity in this specific report we selected arterial hypertension as the most representative cardiovascular risk factor and included it in the demographics table. Given that the distribution of patients with arterial hypertension was very similar in the 3 groups (12.7% vs 8.1% and 13.1%, p=0.229), represented 10.3% of the whole study cohort (48/463) and were overall very well controlled, for space reasons we did not expand further on this topic in this specific report, directed to evaluate the relationship between OCTA and HbA1c levels. We do believe that if this parameter could have had any influence in the OCTA metrics reported, it may have possibly affected equally the 3 groups. All these systemic data will be detailed in subsequent reports of this study.  

A second point is whether they have taken into account separately the systolic and diastolic tension levels with these changes. The authors do not take into account some data that may explain the failure to attend the appointment.

Thanks for highlighting this interesting point. As described in the previous comment, we did collect data about the systolic and diastolic blood pressure. However, no significant associations were observed between these parameters and any OCTA measurements in any of the study groups, and for this reason no adjustments were done for blood pressure in the linear models presented in this report, conversely to other factors that were included in the models (such as age, sex, signal strength index, diabetes mellitus duration and axial length).

However, we have included a mention to the collection of systolic and diastolic blood pressure in the methods section (page 3, line 106), to highlight that this data was also collected, as we agree that this point may be of interest for readers.

2. Another factor that can affect retinal vascularization is renal status. Have the authors taken into account renal function (through glomerular filtration or the presence of albumin in urine) with the findings in the OCTA?

We do agree with the reviewer that the kidney status is a very important factor that deserves specific attention. Indeed, we have prepared a separate report of this study that is currently under review in another journal (Alé-Chilet et al, “OCTA in type 1 DM. Report 2: Diabetic Kidney Disease”) directed to evaluate the association of OCTA parameters and kidney function tests, such as glomerular filtration rate, albumin/creatinine ratio and the international KDIGO classification. For this reason, in the current report we have not included the data related to kidney function.

Minor commentaries 

1. The authors base the study on eyes and not patients, which is correct, but could indicate how many patients are studied in each section of Figure 1.

Many thanks for the comment. We have now created a new Figure 1 to highlight the number of patients and eyes included in each section. We do hope that the number of patients and eyes included in each analysis is clearer now.

2. In the introduction the authors make the following sentence: “The advent of artificial intelligence applications in ophthalmology has made possible to estimate blood HbA1c levels from fundus retinographies based on existing large datasets from DR screening programs,…”.

I think that since it is not proven, they should change it since it implies that through a retinography we can determine the value of HbA1c which is not true.

We do agree with the reviewer that it is not possible to determine the value of HbA1c with a retinography at the present time. The sentence has been rephrased to limit this statement to preliminary studies in labelled datasets, and a specific mention to the poor external validity of these algorithms in independent cohorts has been included.

Resume.

In conclusion, an interesting study but with some questions that the authors should answered

Many thanks again to the reviewer for their comments, that have definitely allowed us to improve the quality of the manuscript.